# T84 Monolayer Cell Cultures Support Productive HBoV and HSV-1 Replication and Enable In Vitro Co-Infection Studies

**DOI:** 10.3390/v16050773

**Published:** 2024-05-13

**Authors:** Swen Soldwedel, Sabrina Demuth, Oliver Schildgen

**Affiliations:** 1Kliniken der Stadt Köln, Institut für Pathologie, 51109 Köln/Cologne, Germany; 2Institut für Pathologie, Klinikum der Privaten Universität Witten/Herdecke, Ostmerheimer Str. 200, 51109 Köln/Cologne, Germany

**Keywords:** HSV-1, HBoV-1, cell line, replication, monolayer

## Abstract

Based on several clinical observations it was hypothesized that herpesviruses may influence the replication of human bocaviruses, the second known parvoviruses that have been confirmed as human pathogens. While several cell lines support the growth of HSV-1, HBoV-1 was exclusively cultivated on air–liquid interface cultures, the latter being a rather complicated, slow, and low throughput system. One of the cell lines are T84 cells, which are derived from the lung metastasis of a colorectal tumor. In this study, we provide evidence that T84 also supports HBoV replication when cultivated as monolayers, while simultaneously being permissive for HSV-1. The cell culture model thus would enable co-infection studies of both viruses and is worth being optimized for high throughput studies with HBoV-1. Additionally, the study provides evidence for a supporting effect of HSV-1 on the replication and packaging of HBoV-1 progeny DNA into DNase-resistant viral particles.

## 1. Introduction

Herpesviruses are widely distributed DNA viruses that are responsible for mild to severe diseases [1,2]. The human Herpes simplex virus type 1 (HSV-1) is mainly responsible for skin lesions at the lips but also can cause life-threatening clinical courses such as herpes encephalitis [3]. On a molecular level, HSV-1 is also known to act as a helper virus for the replication of other viruses such as SV40, and dependoviruses such as the adenovirus-associated virus (AAV).

The dependoviruses belong to the parvoviruses, a group of naked viruses with a single-stranded DNA genome. Among the human parvoviruses are the parvovirus B19, the human Bocavirus (HBoV), and the aforementioned dependoviruses. This latter group of parvoviruses is fully dependent on either adenoviruses as helper viruses or even the HSV-1 DNA synthesis machinery [4]. The parvovirus B19 and HBoV, however, are still genetically and biologically related to the dependoviruses, and despite being autonomous viruses it cannot be excluded that the presence of a helper virus such as a herpesvirus or an adenovirus may synergistically support the replication of autonomous parvoviruses.

In addition to these commonly known facts and the observation by several groups that HBoV is frequently detected as a co-pathogen [5,6], our group has made a series of clinical observations that support the hypothesis that the HBoV replication cycle is influenced by herpesviruses. First, we and others have identified HBoV in up to 25% of colorectal and lung cancers and later also in up to 60% of tonsillar tumors [7,8,9,10,11]. In concert with the fact that HBoV establishes a covalently closed circular DNA [12,13], we assumed that it may persist, which was supported (if not confirmed) by a clinical case in which HBoV cccDNA was detected over more than six months, while in the phases with clinical symptoms of cough the cccDNA was found as a relaxed circle, whereas in the asymptomatic phases the cccDNA was coiled or supercoiled [14]. Those observations resemble the mechanism of development of the hepatitis B virus (HBV) induced hepatocellular carcinoma (HCC) [15], and indeed HBoV is able to trigger procancerogenic cellular pathways in vitro [16].

A further observation we made was that we found HBoV-DNA in two co-infection cases along with herpesviruses in patients with lung fibrosis [17,18]. The development of the HBV-HCC, of note, is also characterized by a stage in which organ fibrosis occurs. This observation was the basis for a further hypothesis of our group. Taking into account that herpesviruses could support dependoviruses, i.e., relatives of HBoV, as helper viruses, and as herpesviruses are able to replicate plasmids with SV40 origins, we assumed that herpesviruses may be responsible for the activation or reactivation of persisting HBoV cccDNA and act as helper viruses even if HBoV is in principle autonomous. This hypothesis, beyond a pure epidemiological explanation, could be the reason for the frequent co-detection of HBoV along with other respiratory viruses.

Based on those thoughts, we aimed to develop a cell culture model in which those hypotheses could be tested. To date, only three cell culture models, namely primary air–liquid interface cultures [19], CuFi-8 air–liquid interface cultures [20], and T84 air–liquid interface cultures were described to support HBoV replication [21]

While we contributed to the very first primary cell culture of HBoV [19] and have implemented a further cell culture model based on those earlier progress and a reverse genetics system [20]), those earlier cell culture models were cost expensive, took a lot of time before cells could be productively infected, and were limited to small experimental series in sense of the overall volumes and the technical limitations by the air–liquid interface cell culture systems. A novel cell line identified as permissive by colleagues Dirk Grimm and Steeve Boulant from Heidelberg and kindly provided to our lab a few years ago are T84 cells [21], which are permissive for HBoV as air–liquid interface cultures but cost less than CuFi8 and primary cells. T84 cells originate from the lung metastasis of a colorectal tumor and form a pleomorphic and rather ugly monolayer, the latter an observation that triggered our idea that even the monolayer could be permissive for HBoV. As shown below, this hypothesis was successfully tested here, and we therefore aimed to initiate a pioneering experiment in which we wanted to test the co-infection potential of HBoV and a human herpesvirus. Although the clinical cases we observed did not include an HBoV/HSV-1 coinfection we decided to use HSV-1 as a putative helper virus. The reasons were rather pragmatic, i.e., HSV-1 is a virus we used in the past to study herpesviral DNA replication, we have access to several conditionally lethal temperature-sensitive variants for future studies, and the virus has a nearly ubiquitous cell tropism, while other human herpesviruses such as HHV-6 from one of the clinical reports have a more narrow cell tropism and are unlikely to infect T84 cells. Consequently, in this study, we combined the resources of our group from “both worlds” and combined the T84 HBoV model with our well-characterized HSV-1 strain to test the hypothesis that T84 is a cell line permissive for both pathogens and that HBoV and HSV-1 may influence each other.

## 2. Materials and Methods

a.Cell cultures and monitoring of cytopathic effects

In a previous report, the cell line T84 (ATCC™ CCL-248, Manassas, VA, USA) was described to be permissive for HBoV if cultivated as air–liquid interface organoids [21]. The cell line is derived from the lung metastasis of a human large intestine colorectal tumor and can easily be grown as monolayers in DMEM/F12 with 5% (*v*/*v*) Fetal Bovine Serum (FBS) and 1% (*v*/*v*) of a cell culture-ready penicillin–streptomycin solution (all Gibco/Thermo Fisher, Waltham, MA, USA).

In order to characterize putative cytopathic effects of HSV-1 and HBoV-1 on the monolayers of T84 cells, we have made use of a novel multisensory real-time cell analysis platform. The CYRIS^®^ system (INCYTOИ^®^ GmbH, Planegg, Germany) monitors and analyses metabolic and cell morphological changes in a fully controllable atmospheric environment as originally published by Lengauer and coworkers [22]. The main rationale why we used the system was its ability to take a controlled series of photographs that enables a video-recording of the cytopathic effect in form of a time-lapse movie composed of single picture taken every 20 to 30 min. As no data were yet available on the metabolic changes and metabolic stress during the cytopathic effect, such as acidification that occurs, e.g., during some intoxication, the additional measurements were a welcome add-on feature for the empiric analyses. The system is not humidified and does not yet enable a fully sterile long-term observation atmosphere, thus bearing the risk of contamination with every door-opening event, the latter hindering frequent media changes and consequently limiting longer observation periods.

Moreover, CYRIS allows a label-free measurement that, depending on the experimental setting, can be maintained for days or weeks, depending on the experimental setup. The sensor plates consist of 24 wells of which each has a defined reaction chamber and attached media inlet and outlet to ensure a continuous supply of culture media (Figure 1A–C). The analysis plate used here measures metabolic parameters (pO_2_ and pH) and has a dedicated microscopy window. The metabolic parameters are measured as oxygen consumption rate (OCR) and extracellular acidification rate (ECAR). T84 cells were grown as described above and transferred to the sensor plates a day before the inoculation with viruses started. Then, an indicator-free CYRIS standard medium was used during the entire observation time that was replaced every 20 min and analyzed with the sensoric measurements. Microscopy images were taken every 20 to 30 min.

b.Virus strains, inoculation and quantification of viral genomes

Virus strains used were HSV-1_17_syn + [23,24] and HBoV-1 [20], the latter generated from a plasmid system as previously described [20]. The generation of HSV-1_17_syn + stocks also was published previously [24]; in brief, Vero cells were kept in DMEM standard medium containing 5%FCS and penicillin and streptomycin and infected with an MOI of 0.1 before cells and supernatant were harvested after full cytopathic effect was observed, generally latest 72 h post-inoculation. The quantification for the present study was performed by qPCR in order to use the same method for genome quantification as for HBoV, as no plaque assay was yet known for HBoV [7,14,25].

A normalization was not necessary as in all cases the same volumes and inocula were used, which was a major advantage of the Cyris assay. Of note, the Cyris system replaces the medium every 20 min with 200 µL fresh medium. The supernatant is collected. Consequently, a total volume of 57.6 mL (96 h, 3 medium changes per hour, i.e., 96 × 3 × 0.2 mL) total volume was collected. Of those volumes, nucleic acid extraction was performed.

Due to the fact that the CYRIS system was provided for a limited pilot testing phase, we used only a single low MOI and were not able to optimize the infection and culturing conditions. This is a major reason why present the data as a preliminary communication. The low MOI was chosen to see the maximal possible difference between the inoculation and the endpoint of the infection. As discussed in one experiment below, it appears that an improvement in the culturing conditions could lead to a serious increase in the monolayer replication efficacy.

Mock infections were performed with exhausted cell culture medium that was incubated with cells for the same time as the virus stock production lasted. This was used as control in order to reduce a bias in the ECAR and OCR measurements (see below). Any inoculations were performed with low MOI of approximately <1. Cell culture infections were performed in quadruplicate on several independent plates as HSV-1 monoinfections, HBoV-1 monoinfections, or double infections with HBoV-1 and HSV-1, respectively. Cell culture supernatants were collected for quantification. As no plaque assay was established for HBoV, all quantifications were performed with qPCR. The qPCR for HBoV was published previously [7,14,25], for HSV-1 the IVD qPCR artus^®^ HSV-1/2 PCR Kit (Qiagen, Hilden, Germany) was used as described in several studies [26,27,28,29].

For the statistical analyses of the PCR results and the differences between the mono- and co-infections of HSV-1 and HBoV, descriptive statistics were applied. First, the 95% confidence interval of the covariance was determined along with the standard deviation and the Pearson coefficient. As the group size (i.e., quadruplicate) is rather low, a Yates correction was applied, too, in the χ^2^-test.

In order to discriminate between infectious particles and free viral nucleic acids from cell debris, infected cells or free excreted DNA, a portion of the supernatant was subject to a rigorous DNase treatment as previously described [19], and the difference between the copy number with and without DNase treatments was determined for each infection.

## 3. Results

T84 cell monolayer cultures were inoculated with MOI 0.1 of HSV-1, HBoV-1, or HBoV-1 plus HSV-1 (double inoculation). Mock infection was performed with an exhausted cell culture medium in order to emulate the effect of cellular debris and metabolic waste that occurs after cytopathic effects. Monolayers were screened microscopically each day and cytopathic effects were analyzed. After 96 h, supernatants were collected and treated with DNAase. As shown in Figure 2, a clear genomic amplification of both HSV-1 and HBoV-1 is observed in T84 cells. The DNase treatment leads to a lower copy number of both HSV-1 and HBoV-1 co-infected and HBoV-1 monoinfected cells. In both cases, however, there is still a clear increase in DNase resistance, i.e., encapsidated progeny DNA. Thereby, the co-inoculation of T84 cells with HBoV-1 and HSV-1 leads to a 1.48-fold higher amount of DNase-resistant HBoV-1 genomes. This correlation of increased HBoV-1 DNA replication in the presence of HSV-1 is statistically significant: In a 95% confidence interval the covariance was 109.59, x-standard deviation 16.94, y-standard deviation 6.47, and a correlation coefficient of 1 (Pearson), and in the χ^2^-test with Yates correction the χ^2^ is 3775.1226 with a *p*-value < 0.00001. The large error bars resulted from several biologically independent experiments, and all experiments were performed in quadruplicate at least. In one experiment, however, the increase in HBoV-DNA in the co-infection was 160 -old instead of 36 to 40 fold in the other experiments.

The long-term observation with the Incyton technology revealed that both HBoV and HSV are able to induce a clear cytopathic effect in T84 cells as shown in the Appendix A and display an oxygen consumption rate and extracellular acidification rate similar to the mock infections, i.e., no significant differences between these parameters were observed (Figure 3).

In contrast to the ECAR and OCR observation that solely revealed a non-significant tendency for an alteration in the infections with HSV-1 versus the mock controls, both HBoV-1 and HSV-1 induced a remarkable cytopathic effect. This effect is shown exemplarily in Figure 4 and is also shown in the Appendix A.

The cytopathic effects (CPE) of both viruses differ from each other. The fact that a CPE is formed during the HBoV infection was surprising, as this was not yet described for monolayer cell cultures. For HSV-1 this was not surprising, as virtually all permissive cell lines suffer from a CPE after the HSV-1 infection.

While HSV-1 induces a classical focal lesion with cells that increase in their volume before they become round bubbles detaching from the monolayer, HBoV induces a diffuse and embracing cytopathic effect with cells becoming rather small and reorganizing in several layers before detaching from the ground (Figure 4 and Appendix A). In double infections, the cytopathic effect is characterized by a mix indicating a local dominance of either virus.

## 4. Discussion

Our data indicate that T84 cells are permissive to HSV-1 and also are able to support the replication cycle of HBoV-1 even if the T84 cells are cultivated as a monolayer culture and are not differentiated to air–liquid interface cultures as previously described [21]. This information is important as it would simplify studies on the replication of HBoV-1 and could also be used, e.g., to isolate HBoV wild-type strains. However, our study is solely a proof of principle, and further efforts are needed, as the replication efficacy measured in encapsidated progeny HBoV-1 is still rather moderate.

The DNA replication of HBoV is moderately enhanced by HSV-1, i.e., there is a slight but statistically significant increase in the formation of encapsidated HBoV-progeny DNA compared to the single infections. This observation supports our initial hypothesis that there could be an interaction between the two viruses as seen between dependoviruses and adenoviruses and/or herpesviruses. This interaction would also explain the putative reactivation of HBoV from a persisting stage and the frequent co-detection with herpesviruses as, e.g., observed in some clinical cases. However, although the preliminary results observed here are promising, the overall replication of HBoV in the monolayer culture is low compared to other pathogens that replicate in an exponential manner. Thus, future studies are required to elucidate which factors hinder a more rapid and more productive replication cycle in vitro, e.g., by comparing the transcriptomes of air–liquid interface-grown T84 cells versus their monolayers in presence or absence of HBoV. Also, the role of HSV-1 as a cofactor, especially of HSV-1 DNA replication enzymes, could be further analyzed, e.g., by the usage of HSV-1 trains with temperature-sensitive lesions [30].

As a “side-effect” we have observed that the ECAR and the OCR are not affected by the HSV-1 and/or the HBoV infection. This result, however, is not less important, as it appears that the metabolic functions of the cells remain stable even if a viral infection occurs, and the cytopathic effect induced by HBoV and/or HSV differs biochemically from many pharmacological effects, at least during the phases we measured.

The measurements of OCR and ECAR could have been longer along with a longer observation of the development of the cytopathic effects. This missed opportunity comes from technical limitations, We made use of an early version of the CYRIS system, that did not allow full sterile culturing during the entire observation period and required the addition of a new medium after 72 to 96 h, at least in our experimental setting. We used these settings, however, to analyze whether the progeny viruses were really shed and “washed” away, which was the case as the quantifications were made from the “waste”, i.e., the supernatants that were washed away every 20 min and replaced by fresh, virus-free medium. This experimental setup, however, underlines that true viral shedding and release of encapsidated viral DNA especially of HBoV occurred. The lack of an effect of the cytopathic effects on the OCR and ECAR could be caused by the fact that the time between two washing steps was too short to measure a CPE-associated metabolic effect, whereas the pH-dependent color change in standard culturing media during monolayer cultures is a cumulative effect in cell culture flasks.

It must also be discussed that in the co-infection setting a single experiment led to a 160-fold increase in HBoV DNA compared to lower increases in the other experiments, leading to large error bars. This, however, may be explained by the hypothesis that the optimal culturing conditions in monolayers are not yet fully elucidated and warrant further research and funding of this topic.

A further limitation of our preliminary communication study is that the cytopathic effects were characterized solely by bright field serial microscopy. Future attempts should also include immunochemical staining, preferable and provided the availability, with antibodies against both viruses. As discussed during the peer review, those further studies also should include other co-infections with common viruses such as RSV, HMPV, coronaviruses, and adenoviruses, which also could have an impact on HBoV replication.

In summary, the data support the hypothesis of an interaction between HBoV-1 [7,8,16] and human herpesviruses and would explain the frequent co-detections by reactivation of persisting HBoV by these interactions.

## Figures and Tables

**Figure 1 viruses-16-00773-f001:**
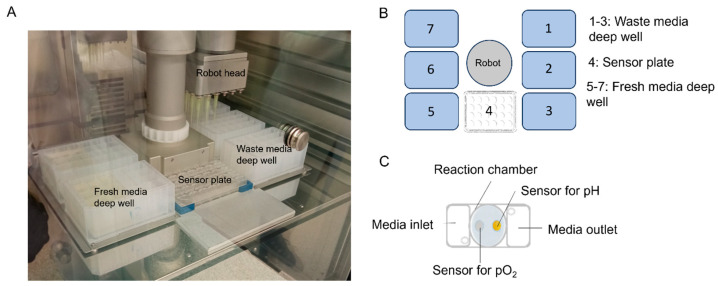
**Setup of the CYRIS cell analysis platform.** (**A**) Photo of the robot working space. (**B**) Scheme of the spatial arrangement of the working space. Deep wells containing fresh medium are placed on the left side (5–7), the sensor plate is placed up front in the middle (4) and deep wells for waste medium are placed on the right side (1–3) of the area. Medium is removed from the sensor plate through the respective outlets and transferred to the corresponding waste medium deep wells, followed by addition of fresh medium from the corresponding reservoir into the sensor plate inlets. (**C**) Scheme of one well. Media inlet and outlet are connected to the reaction chamber creating a media flow along the reaction chamber. The figure was created using Microsoft PowerPoint (Version 2306) and Servier Medical Art (Creative Attribution 3.0 France).

**Figure 2 viruses-16-00773-f002:**
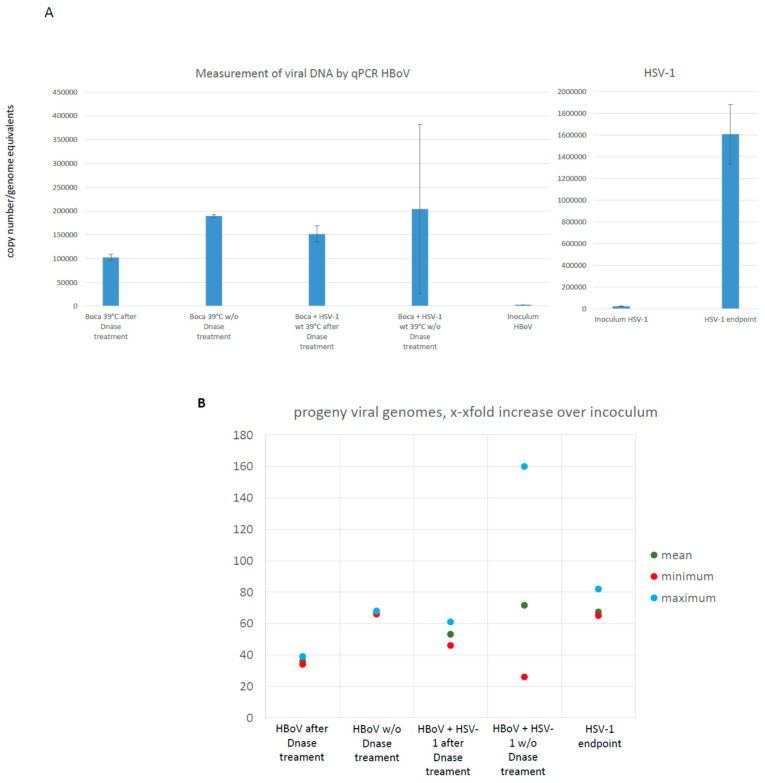
**Results of HBoV-1 qPCR with and without DNase treatment.** The figure shows the results of mono- and double-inoculation experiments of HBoV and HBoV/HSV in T84 monolayer cultures. DNase treatment, not surprisingly, leads to a reduction of amplified genome equivalents but shows that a remarkable and measurable production of encapsidated viral progeny genomes was generated, i.e., a productive infection with viral particles released to the cell culture supernatant has occurred. The presence of HSV-1 has a positive effect on the HBoV-1 DNA replication and increases the output. This increase is statistically significant (see main text). In part (**A**) the copy numbers as measured by qPCR are given, while part (**B**) shows the relative increase compared to the inoculum, i.e., the ration between the final endpoint of the experiment and the inoculum is shown. A positive value thereby indicates successful DNA replication and in case of DNase-resistant DNA also encapsidation.

**Figure 3 viruses-16-00773-f003:**
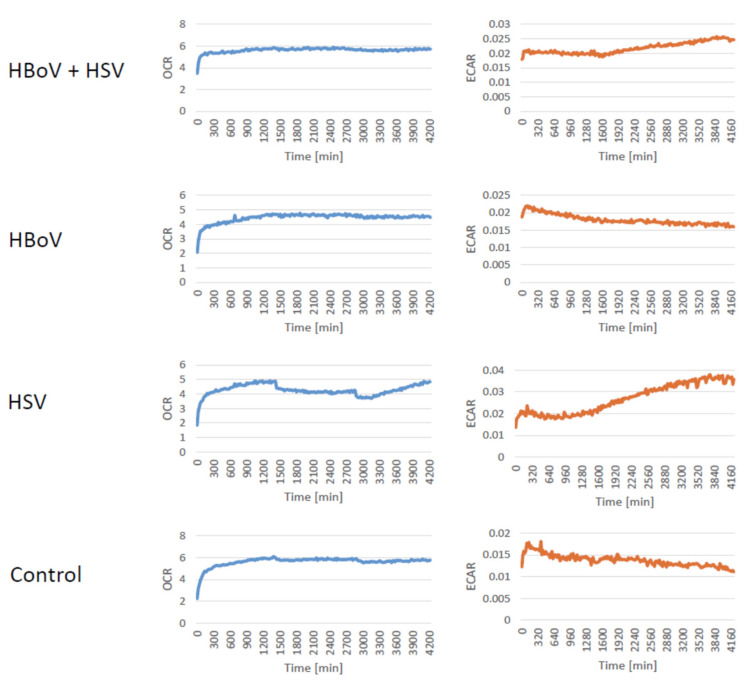
Oxygen consumption rates and extracellular acidification rates of representative long-term observations of HBoV and HSV-1 mono- and coinfections of T84 cell monolayers kept in the Incyton incubator. Despite clear cytopathic effects (Figure 4, Appendix A) no significant differences could be observed. A tendency for a decrease in the ECAR is visible for infections in which HSV-1 is present.

**Figure 4 viruses-16-00773-f004:**
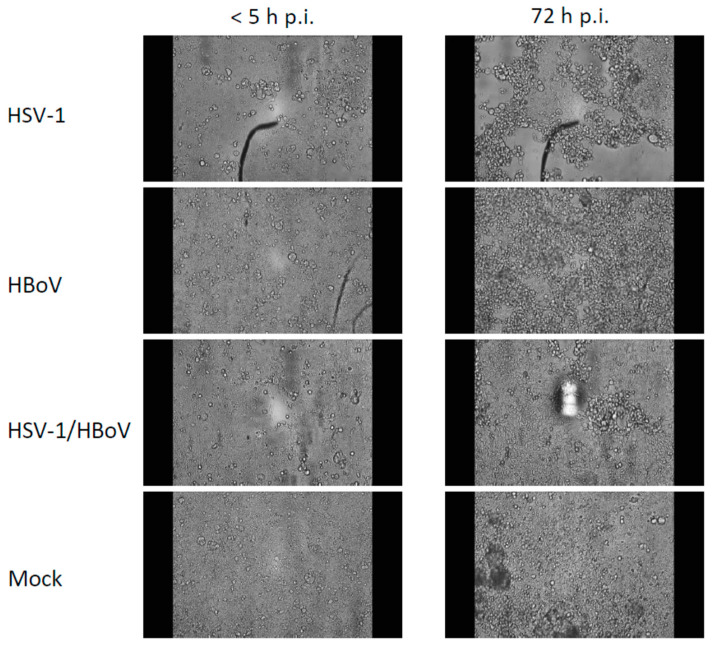
Cytopathic effects of HBoV, HSV-1, and HBoV/HSV-1 coinfections of T84 cells. The photographs are taken from the Incyton monitoring series and represent single time points. The respective videos are attached as Appendix A. HSV-1 cytopathic effects are classical focal lesions with increasing diameter and cells detaching from the monolayer. The HBoV cytopathic effect is rather diffuse and affects the entire monolayer. In double infections, the cytopathic effect is a mixture between the single-infection types of cytopathic effects and is less pronounced. The Appendix A show the development of the CPE.

## Data Availability

All relevant data are included in this manuscript and the Appendix A.

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
