# Peer review of "T84 Monolayer Cell Cultures Support Productive HBoV and HSV-1 Replication and Enable In Vitro Co-Infection Studies"

_viruses, 2024, doi:10.3390/v16050773_

Round 1
Reviewer 1 Report (Previous Reviewer 2)
Comments and Suggestions for Authors
After reading several statements from other reviewers, I have to agree with many points made. I see the authors describe the manuscript as "preliminary data" and have submitted said manuscript as a 'communication' however I do not see where this is an option for article type under journal overview?
If this is an option for submission, the article in present form is valid for publication. As an 'article' however, several key issues need addressed prior to publication. As such, if no 'communication' feature is offered, perhaps the authors should consider submitting the article as a pre-print?
Author Response
Dear Reviewer,
many thanks again for your valuable comments. We agree that some data need future work and additional efforts. The journal allows the communication format, thus we decide to keep the communication format.
We acknowledge that the manuscript has significantly improved by your comments and suggestions, and will take them into account for future work.
Best regards
Oliver Schildgen
Reviewer 2 Report (Previous Reviewer 1)
Comments and Suggestions for Authors
Commend the authors extensive improvement of the text portion of this preliminary communication. The only suggestion is the inclusion of error bars in Fig. 2B.
Comments on the Quality of English Language
For the written English, specific attention should be noted for the correct use of specific words, spelling, and ease of reading. Editorial assistance may be beneficial to ensure clear communication of these results.
Author Response
Dear Reviewer,
we thank you very much for the important comment. As it is difficult to provide a real error bar for the x-fold change, we have changed the graph format and have included a minimum and a maximum fold change and the mean, which also explains the large error bars in the other graph format and which was the most honest way to present the data.
We hope you agree with this idea that was triggered by your comment.
We also would like to repeat that we are thankful for the valuable comments and critical suggestions that improved our manuscript.
Best regards
Oliver Schildgen
This manuscript is a resubmission of an earlier submission. The following is a list of the peer review reports and author responses from that submission.
Round 1
Reviewer 1 Report
Comments and Suggestions for Authors
Human herpes simplex viruses (HSV-1 and HSV-2) are ubiquitous neurotrophic pathogens for which latency-reactivation cycles are a continued source of viral shedding and disease burden. At the same time human bocavirus 1 (HBoV-1) is widely detected in the nasopharyngeal samples during respiratory tract illness.
Schildgen et al. evaluated the “interaction” between the coinfection of a monolayer of T84 cells with HboV-1 and HSV-1. An experiment evaluated HBoV viral shedding levels, oxygen consumption, and the acidification of the media during a 96-hour infection.
As detailed below, comments are provided to clarify experiment details and conclusions drawn from these results. Additional experiments are strongly suggested to further support the conclusion that “HSV-1 has a positive effect on HBoV-1 DNA replication” but also an attempt to address the possible mechanism for this observation.
Specific comments:
-Lines 33-39: Based on the references provided only the Streiter el al. 2011 article identified one patient with HHV-6 and HBoV as opposed to “two coinfections”. Also, unclear why the remaining three references are provided for this passage by which some of the patients are negative for other pathogens outside of HBoV. Given this clinical observation, it would also be informative if it was discussed why HSV-1 was used and not HHV-6A/B for this study.
-Lines 50-66, Figs. 1, 3: There is no attempt to make clear or the purpose of why the CYRIS system was utilized to collect pO2 and pH levels during HBoV and HSV-1 infection. Given there was no significant difference during the course of the 96-hour infection not sure what these figures add to the manuscript.
-Lines 78-80: What was the methodology of generating HSV-1 viral stocks?
-Lines 77-89, Fig. 2: Why was the endpoint for the infections stopped at 96 hours? Seems a missed opportunity to evaluate kinetics of viral shedding over multiple time points.
-Lines 87-89, Fig. 2: HSV-1 DNA copy numbers are not provided. This is the only direct proof (required positive control) that the cells were productively infected with HSV-1. Brightfield images of cells undergoing cytopathic effect is not sufficient unless immunofluorescence-based detection for HSV-1 viral antigen is included.
-Fig 2, Lines 100-102: Results do not support the conclusion that “clear genomic amplification of both HSV-1 and HBoV-1”. Based on the figure title and legend, not clear if the viral DNA copy levels is for HBoV or HSV-1.
-Fig 2, Lines 102-110: With the very large error bars and only a “1.48-fold higher amount of DNase resistant HBoV-1 genomes” its surprising these results were considered “statistically significant.” Furthermore, there is no indication how many biological (not technical) experiments were completed to generate these results. Given this the results do not support the conclusions “the presence of HSV-1 has a positive effect on the HBoV-1 DNA replication and increases output. This increase is statically significant.”
-Fig 4, Lines 131-135, 143-148: Is it surprising that HSV-1 and HBoV infection results in CPE? Not clear what this anticipated observation adds to the manuscript.
-Lines 85-94, Fig 2: How is the data normalized to account for differences in volume between samples and what is the sensitivity of the qPCR assays? Further details in the method are requested (reagents used, primer sequences, controls, instrument).
-Lines 159-161: Collectively the results in Figure 2 do not support the conclusions that “these observations supports our initial hypothesis that there could be an interaction between the two viruses…” As detailed above, difficult to consider 1.48 fold increase as significant given the large error bars with no indication of number of replicate experiments. Moreover, the use of “interaction” is an ambiguous conclusion. Additional experiments should strongly be considered to not only characterize this “interaction” as well as begin to address the mechanism behind the possible replication advantage of HBoV with HSV-1. Some examples include evaluating the kinetics of viral shedding, use of recombinant HSV-1 viruses to address at what stage or viral protein is involved, etc…
-One very critical experiment is to demonstrate that a single cell is coinfected with HBoV-1 and HSV-1. Assays such as immunofluorescence images paired with counting of a random collection of cells (from at least three biological experiments, not technical) for the % of cells coinfected. Also, are there any thoughts or results for the replication of HBoV-1 on monolayer as compared to air-liquid interface cultures of T84 cells?
Editorial comments:
-Lines 29-33: Would be advantageous to the reader if more details are provided regarding “on the molecular level HSV-1 is also known as a helper virus for replication of other viruses”. Expanding the introduction will provide a greater foundation for the study presented.
-Supplementary video files: Recommended the authors provide several time stamp indicators (12 hour, 24 hours, etc) as a point of reference by which the rate of CPE is occurring.
Reviewer 2 Report
Comments and Suggestions for Authors
In the article by Soldwedel et al, the authors describe a novel approach to studying human bocaviruses/HSV-1 coinfections. This study found that T84 cells support human bocavirus replication when cultivated as monolayers rather than the traditional air/liquid interface organoids. Using the multisensory real time cell analysis platform (CYRIS), the authors claim that 96 h post-infection, viral DNA is increased and that bocavirus DNA is higher when co-infected with HSV-1. Interestingly, cellular acidification and oxygen consumption was not increased during co-infections compared to single- or mock-infected wells. The authors use this and clinical data to state HSV-1 may enhance the replication and packaging of human bocavirus DNA.
This short communication is clear and succinct. The authors use of a novel technique is highly applicable and a benefit to the field of difficult-to-culture viruses. Listed below are some considerations for the authors:
1. Several different abbreviations for human bocaviruses is used including HBoC, Boca, and HBV; please choose one and remain consistent throughout.
2. The authors state HBoC is "a dependovirus-relative pathogen," please expand upon this i.e. more details as to the difficulty of in vitro culturing, in vivo models etc.
2a. Further, HBoC is included as a mono-infection, co-infection with HSV-1, and uninfected controls; does there exist a 'positive' control such as HBoC co-infections with RSV, human rhinovirus, or another standard model? Could the data be compared to the HAE-ALI model?
3. Some more details in the methods section would be useful:
3a. How did the authors choose this MOI? Were different MOIs tested? Were the viruses added at the same time? Were different infection times tested (infecting with one virus first for X hours and adding second later etc)?
3b. The authors state the qPCR assay for HBoC was previously published however no citation is included.
3c. Additional details for the CYRIS and Qiagen IVD qPCR for HSV-1/2 are requested or a citation outside the manufacturer info (as their website provides little beyond press release). What does an OCR of 5 mean? Are the values averaged for replicates? Are statistics calculated?
3d. Please give more specific details for the calculations of statistics; move the info from lines 107-110 here.
3e. The experiments were done in quadruplicate wells, however was more than one plate performed? Is there concern for 'edge effects' from the plates? is the CYRIS chamber humidified?
3f. Details of supplemental video capture are absent; please include.
4. Converting the graph in Figure 2 to fold-change might improve ability to differentiate changes between single and co-infection results. Further, the error bar from co-infection w/o DNAse treatment leads to skepticism of statistical difference. Do the authors have an explanation for this substantial variation? Please include asterisks indicating significance and which data points are being compared.
5. In the supplemental videos, the HSV-1 only video show ~75% confluence of monolayer compared to the other 3 (mock, HBoC, and co-infection). Were cells counted and plated equally among wells? Could differences in "cytopathic effect" simply be the monolayer aging and dying from overcrowding?
6. Discussion: The authors show substantial CPE from the MP4 files however the CYRIS does not show changes is ECAR or OCR; can the authors expand on why this might be?